# OpenReview forum: "Introspective Planning: Aligning Robots' Uncertainty with Inherent Task Ambiguity"
_NeurIPS.cc/2024/Conference — NeurIPS 2024 poster_

### Official Review · Reviewer_RRQ3 · 2024-07-13

**Soundness:** 3
**Presentation:** 4
**Contribution:** 4
**Rating:** 7
**Confidence:** 1

**Summary:**

The paper explores introspective planning to enhance robotic task execution using large language models (LLMs). The authors introduce a method for LLMs to form uncertainty-aware plans without fine-tuning while addressing hallucination and task ambiguity. Their approach integrates introspective planning with conformal prediction.

**Strengths:**

The use of introspective planning with LLMs.
The write-up is easy to follow.
Sufficient comparison and ablation experiments.

**Weaknesses:**

NA

**Questions:**

NA

**Limitations:**

The authors addressed the limitations of their proposed work.

---

> ### Author Rebuttal · Authors · 2024-08-07
>
> We thank the reviewer for the positive feedback, acknowledging that our study includes sufficient comparison and ablation experiments. We also thank you for the comment that the write-up is easy to follow. We are looking forward to receiving additional comments and feedback from you during the discussion phase.

---

### Official Review · Reviewer_euid · 2024-07-13

**Soundness:** 2
**Presentation:** 3
**Contribution:** 2
**Rating:** 3
**Confidence:** 5

**Summary:**

This paper introduces a method that uses introspective planning to guide LLMs in forming uncertainty and ambiguity-aware task plans. The proposed method derives and quantifies the inference uncertainty of LLMs to enhance task planning and conformal prediction. Additionally, a new dataset on safe mobile manipulation is created as part of this work.

**Strengths:**

+ Leveraging LLMs to improve robot task planning performance is promising.
+ The concept of introspection is interesting.
+ The created dataset on Safe Mobile Manipulation benchmark could benefit the robotics community.

**Weaknesses:**

- The term "introspection" or "introspective" is not scientifically defined. Additionally, does introspective refer to a robotic agent, the LLM, or the proposed approach?

- The novelty of the paper is unclear, especially compared to [31]. Line 63 states "The fundamental aim of introspective planning is to guide LLMs automatically to reason about task uncertainty and assess the feasibility of candidate actions." [31] also reasons about uncertainty and ambiguity through MCQA. Is [31] introspective?

- The paragraph in Line 135 indicates the significant enhancement is from the introspective planning rationale k_i, and Line 48 indicates that introspective planning provides a tighter statistical guarantee. However, no theoretical or mathematical proofs are provided.

- How does the proposed approach enable the new capability of modeling and addressing task safety? The method of handling safety seems identical to addressing ambiguity through MCQA.

- What robotics simulations or physical robots are used to create the safe mobile manipulator dataset?

**Questions:**

- Refer to the comments in the Weaknesses section.

**Limitations:**

No negative societal impact of the work is perceived.

---

> ### Author Rebuttal · Authors · 2024-08-06
>
> We thank the reviewer for the thoughtful feedback!
>
> > *The term "introspection" is not scientifically defined. Does introspective refer to a robotic agent, the LLM, or the proposed approach?*
>
> We use the term “introspection” as defined in  [19], referring to the human ability to assess internal values and knowledge of abilities to guide domain-level reasoning. In our paper, we extend the usage by analogy to non-human agents—and in particular to language-enabled AI agents—and denote by “introspective planning” a decision-making procedure by which these agents “assess their own confidence regarding task compliance and safety for multiple candidate plans”.
>
> The term “introspective” refers to the planning procedure followed by the agent. In our paper, this planning procedure relies on a large language model, but it is not identical to it. In fact, our planning procedure is introspective regardless of what LLM it uses. The LLM itself (e.g., GPT-4) is not intrinsically introspective but is queried in an introspective manner.
>
> > *What is the novelty compared to [31]? [31] also reasons about uncertainty and ambiguity through MCQA. Is [31] introspective?*
>
> In [31], the LLM *does not in fact reason* about its own uncertainty, that is, it does not form any intelligible logical discourse about it. Instead, MCQA is merely reading out the final-layer activations of the language model, which the system designers (and not the LLM itself) then interpret as log-probability estimates (but these do not readily encode well-calibrated probabilities, hence the need for the conformal prediction calibration phase). In contrast with [31], our approach prompts the LLM to explicitly discuss, in natural language, its own uncertainty regarding the task compliance and safety of candidate options, and the generated discussion is then used to inform the MCQA readout. Our experiments show that this procedure drastically improves the planner's performance across multiple metrics.
>
> We note that while our contributions are briefly alluded to at the start of Section 2 (line 63), they are actually outlined in more detail in the Introduction (lines 36–61). We thank the reviewer for their question, and we will emphasize the central importance of our method, which prompts the LLM using retrieval augmentation to generate explicit reasoning in natural language prior to the MCQA query to refine the uncertainty of the resulting judgment.
>
> > *Line 48 indicates introspective planning provides a tighter statistical guarantee. However, no theoretical or mathematical proofs are provided.*
>
> We clarify that our claim regarding our method’s ability to provide a tighter statistical guarantee is empirical, not theoretical (the latter would require us to make theoretical proofs involving the outputs of LLMs, which would be beyond the scope of our paper).
> In particular, we provide a comprehensive benchmark comparison between our proposed introspective–conformal method and various baselines, including the previous state-of-the-art in conformal planning [31]. Our results show significant improvement in balancing task success rate and the robot's decision conservativeness.
>
> Table 1 shows our method's performance with an 85% target success rate, achieving an over-asking rate of about 6%, compared to 51% for the baseline [31]. This means that the baseline [31] is inappropriately asking the user to provide clarification in 1 out of every 2 requests that are in fact unambiguous, while our method does so in just 1 out of every 16.
>
> Similar improvements are evident in Fig. 4g.  *For any target success rate between 70% and 99%, our method achieves a significantly lower over-ask rate; and, equivalently, for any acceptable over-ask rate, our method can provide a higher statistical bound on success rate.*
>
> The latter result substantiates our empirical claim: if it is allowable for the robot to ask for unnecessary clarification in only 1 in every 2 requests, then the strongest statistical guarantee achievable through prior conformal planning [31] is 85% success rate, whereas with our method it is 99%. We will emphasize this more strongly in the final version of the paper to clearly justify our empirical claim about the tightness of statistical guarantees.
> > *How does the proposed approach enable the new capability of modeling and addressing task safety?*
>
> Our approach introduces several key advancements for modeling and addressing task safety:
>
> - We generate safety-relevant examples for the knowledge base (lines 197-203, 526-540).
> - Our approach relies on human-provided valid options to guide the LLM in reasoning about compliance and safety, trusting that human label providers can account for safety when relevant. In Figure 1, we show how these human-provided labels incorporate safety considerations, allowing LLM to generate safety-aware reasoning (e.g., deducing that plastic is unsafe for direct heating on the cooktop).
> - We explicitly prompt the LLM to consider safety when forming its introspective judgments at runtime. The prompt for generating the safety-aware knowledge base is provided in lines 653-654 and Table 9, and the knowledge retrieval prompt for inference is provided in Table 12.
>
> We present qualitative results in Figures 3 and 10 to further illustrate the differences between [31] and our approach in addressing safety.
> > *What robotics simulations or physical robots are used to create the safe mobile manipulator dataset?*
>
> Both our approach and KnowNo focus on high-level planning through human-labeled mobile manipulation scenarios. The creation of the original Mobile Manipulation, TableTop Rearrangement, and our safe mobile manipulation datasets does not rely on simulations or physical robots. However, these datasets can be utilized for physical experiments. Our approach can be readily combined with the framework in [1] that employs physical robots to execute LLM-chosen plans, although doing so is beyond the scope of our paper.

---

### Official Review · Reviewer_QHHV · 2024-07-15

**Soundness:** 3
**Presentation:** 3
**Contribution:** 2
**Rating:** 5
**Confidence:** 4

**Summary:**

The paper tackles the uncertainty quantification problem for task planning with LLMs. Specifically, the paper proposes to first construct a knowledge base using LLM, that contains human-in-the-loop correction and LLM summarization / reflection. Then this knowledge base is used during inference to provide relevant examples in a retrieval-augmented generation (RAG) fashion. Finally, this is combined with conformal prediction to either predict the next action step or ask for clarification due to high uncertainty. Evaluations are performed on a set of text-based task planning datasets and demonstrate improved performance compared to various baselines on multiple metrics.

**Strengths:**

- Uncertainty quantification is an important topic in the context of task planning with LLMs, and the proposed method shows improvement on most of the metrics compared to prior works (and I also find the metrics to be reasonably constructed)
- The paper overall is well-written and easy to follow. Figures are intuitive and helpful for understanding the core contributions.

**Weaknesses:**

- Despite the improvement, the significance of the contribution is slightly unclear when compared to prior work “KnowNo” - it seems that the only difference is that there is an additional “chain-of-thought style LLM summarization” step for the knowledge base and the calibration dataset. Although it is intuitive that additional improvement can usually be gained by chain-of-thought, it remains unclear if the gain is marginal when there is better underlying LLM.

**Questions:**

See "weaknesses" section above.

**Limitations:**

The limitations are described in the paper.

---

> ### Author Rebuttal · Authors · 2024-08-07
>
> We thank you for the feedback, acknowledging that our proposed method shows significant improvement compared to prior works in terms of addressing uncertainty alignment. We also thank you for the question and for providing an opportunity to clarify our contribution and its significance.
>
> > *What’s the difference between introspective planning and Chain of Thought? Why is the contribution significant?*
>
> Our approach does indeed utilize Chain of Thought, but it is significantly more advanced. We observed that simply applying Chain of Thought does not effectively address hallucinations and can often lead to overconfident responses. For example, in Table 1, the overstepping rate for Prompt Set + CoT is 30.8%, while our approach achieves a much lower rate of 3.8%.  Introspective planning which utilizes retrieval augmentation effectively prompts language-enabled agents to proactively refine their own confidence regarding task compliance and safety for multiple candidate plans, with a guaranteed probability that the agent will either execute the actions desired by the user or ask an appropriate follow-up question to disambiguate the user’s intent (lines 52-55).
>
> One of our core contributions is the construction of the knowledge base. We introduce a new, weakly supervised offline knowledge base construction method that guides the LLM to generate human-aligned introspective reasoning examples as post-hoc rationalizations of human-selected safe and compliant plans. This contribution is discussed in lines 39-42, 56-58 of the paper. As a result, our approach guides the LLM to generate more precise plans, as evidenced by state-of-the-art performance on three datasets across different domains with various metrics.
>
> Additionally, we have created a new Safe Mobile Manipulation benchmark, which enhances previous mobile manipulation datasets by including safety-critical scenarios and introduces new metrics (ESR, NCR, UCR, OAR, OSR, UR) to evaluate a planner’s performance in terms of compliance, safety, and degree of conservatives. Our approach effectively reasons about both compliance and safety, achieving state-of-the-art performance.
>
> > *Is the gain marginal when there is better underlying LLM?*
>
> We conducted experiments using both GPT-3.5 and GPT-4, and the results are presented in Tables 1-7 and Figures 4-7. Our observations indicate that introspective planning significantly improves performance regardless of whether we use a stronger model (GPT-4) or a weaker model (GPT-3.5).
> Introspective planning leverages the reasoning capabilities of the language model to achieve superior performance. Therefore, as the underlying LLM becomes more powerful, the potential for introspective planning to enhance performance increases.

---

> > ### Comment · Reviewer_QHHV · 2024-08-13
> > **Response**
> >
> > Thank you for the response -- it has addressed my questions, and I'm inclined to keep my rating.

---

> > > ### Author Response · Authors · 2024-08-14
> > >
> > > Thanks for your response! We appreciate your review and feedback!

---

### Official Review · Reviewer_Gr1N · 2024-07-24

**Soundness:** 3
**Presentation:** 3
**Contribution:** 2
**Rating:** 6
**Confidence:** 4

**Summary:**

This paper presents "introspective planning" as a method to enhance the reliability and safety of robotic task planning using large language models. The proposed method uses introspective reasoning to address uncertainty through a knowledge base consisting of sets of tasks, observations, and candidate plans, along with a rationale for plans to better align with user intent. Additionally, the authors introduce the Safe Mobile Manipulation benchmark with safety-critical scenarios and new evaluation metrics.

**Strengths:**

1. The ability to generate uncertainty over robot tasks, especially in cases of unsafe operations, is particularly important.
2. The paper is well motivated in aligning LLM-generated tasks with the true user intent, given possibly ambiguous inputs.
3. The knowledge base includes a rationale for each task which expands on the KnowNo framework for generating confidence scores.
4. The new additional safety benchmark adds important safety scenarios to existing datasets to better evaluate planning under uncertainty systems.
5. The method is described well, and the availability of the code gives the opportunity for the broader community to build on this work.

**Weaknesses:**

1. The core components of the work are well established. In the proposed direct method, the incremental improvements to existing work are primarily the introspective approach where the LLM is used to generate a rationale for the plan. The conformal prediction method additionally uses a knowledge base to produce statistical guarantees about the prediction which is very important for safe operation. However, the results show significant performance gaps between the direct and conformal prediction. Given this, it would be good to provide a robust analysis of this tradeoff in a general setting.
2. The paper focuses primarily on manipulation tasks that involve somewhat ambiguous items in a kitchen setting such as disambiguating between two sodas in the scene or having the knowledge that a plastic object shouldn’t go into an oven. It would be helpful to include other domains to better show the generalizability of the method.
3. The knowledge base is a key component of the conformal prediction. The authors compare different sizes of knowledge bases, but little is given towards how these should be constructed for a given task and domain. More details on the variations that a user needs to generate should be given. It seems the user needs to be well aware of the failure mode of the tasks to generate an appropriate knowledge base.
4. There may be bias in the user-generated knowledge base, especially for multiple users. It would be good to show how these affect the performance of the system. Perhaps evaluating a system with multiple users without sharing the intended goals would show how well the system aligns across multiple users and a single knowledge base.

**Questions:**

1. It wasn’t initially clear to me whether the knowledge base contains the explanation. Are these stored after construction - they are not in the data files in the code repo.
2. The experimental exploration of the knowledge-base size contained in the appendix is appreciated. Is there a reason that the other metrics were not included in this evaluation? Do you have insights into why the performance drops with the largest knowledge base and what was the variation in tasks contained in the knowledge base?
3. What is the result when the user provides an ambiguous answer when the system seeks clarification?

**Limitations:**

1. As the authors point out, one limitation is that the conformal prediction seems to perform significantly worse than direct prediction. Particularly interesting is that the Unsafe Rate is worse for conformal prediction than direct, which might raise the question as to how much the uncertainty measurement is contributing to safer outcomes.
2. The method is evaluated on a limited diverse set of tasks in a kitchen manipulation setting.
3. The authors make a good observation about multi-label prediction and more investigation is needed to understand the limits around truly ambiguous tasks.

---

> ### Author Rebuttal · Authors · 2024-08-07
>
> We thank the reviewer for the positive and thoughtful feedback!
>
> > *Why are there significant performance gaps between the direct and conformal predictions and what is the trade-off?*
>
> We thank the reviewer for pointing this out. We discussed the trade-off between direct and conformal prediction in lines 219-225. While direct prediction provides more accurate predictions, it cannot guarantee success. Conformal prediction can guarantee success but performs worse than direct prediction. We acknowledged this limitation in lines 318-319. We hypothesized that the performance gap between direct and conformal prediction could be due to the misalignment between the first-token probabilities (conformal prediction) and text answers (direct prediction) [2, 3], but this still requires extensive additional analysis. We agree that understanding and analyzing this performance gap more deeply is crucial and future work should aim to reduce it.
>
> > *The paper focuses primarily on manipulation tasks that involve somewhat ambiguous items in a kitchen setting. It would be helpful to include other domains to better show the generalizability of the method.*
>
> In addition to mobile manipulation and safe mobile manipulation tasks, we also conducted experiments on the Tabletop Rearrangement domain. This task involves moving colored blocks and bowls on a table according to specific instructions. These instructions are designed to include ambiguities in attributes (such as alternative names for objects and colors), numbers (using vague terms for quantities), and spatial relationships (using general terms for directions and orientations). The experimental results for this dataset are provided in Appendix A, and a detailed description of this dataset is in Appendix B.
>
> > *What should the human provide to generate our knowledge base?*
>
> To construct the knowledge base, we query the LLM to generate a set of candidate plans, conditioned on the task, the observation, and hand-crafted few-shot examples. The user then needs to provide all the valid options $\mathcal{G}_i$​ (alphabetic labels such as A, B, C, D). We prompt the LLM to produce rationales k_i based on these labels. Specifically, we use in-context learning with few-shot examples to guide the LLM in generating explanations of why certain options are valid according to the ground truth.
> We provided a detailed explanation of constructing the knowledge base in lines 74-85 and showed an example in Figure 1.
>
> > *Should users be aware of the ambiguity and safety?*
>
> Yes, users should be aware of ambiguity and safety when providing the labels. This awareness ensures that our approach guides the LLM to generate uncertainty- and safety-aware reasoning, resulting in predictions that are both safe and compliant with the user’s request.
>
> > *There may be bias in the user-generated knowledge base, especially for multiple users. It would be good to show how these affect the performance of the system. Perhaps evaluating a system with multiple users without sharing the intended goals would show how well the system aligns across multiple users and a single knowledge base.*
>
> We agree that reducing bias in the knowledge base is important. With less bias, we can use a smaller knowledge base to achieve high performance. When collecting data for the knowledge base, we ensured that users were aware of ambiguity and safety and had varied intended goals, thus maintaining high knowledge quality. Consequently, during inference, we evaluated the performance with users having different intended goals, and the results show that our system can effectively retrieve relevant and helpful knowledge to guide the LLM in reasoning about compliance and safety.
>
> > *Does the knowledge base contain the explanation? Are these stored after construction?*
>
> Yes, the knowledge base contains the explanations  $k_i$, which are stored after construction. These explanations serve as important introspective reasoning examples that guide the LLM to generate more human-aligned reasoning. This procedure is described in lines 76-85 and Algorithm 1. We will emphasize this point in the final version of the paper.
>
> > *Is there a reason that the other metrics were not included in the evaluation of performance vs knowledge bas size? Why does the performance drop with the largest knowledge base and what was the variation in tasks contained in the knowledge base?*
>
> We thank the reviewer for acknowledging our studies in Appendix C. The goal of this section is to show how the size of the knowledge base influences performance. Among all the evaluation metrics, Exact Set Rate (ESR),  which evaluates the model’s ability to generate precise responses, and Success Rate (SR) are the most representative and clearest for supporting our conclusions. We added a plot with additional metric results (OAR, OSR, NCR) on Mobile Manipulation in the pdf.
>
> We evaluated performance on Mobile Manipulation and Safe Mobile Manipulation. With the maximum knowledge base size, there is a slight performance decrease in Mobile Manipulation, while performance remains almost the same for Safe Mobile Manipulation. We hypothesize that this slight decrease could be due to shortcomings in the retrieval process. Although our approach retrieves examples relevant to the target instruction, it does not always guarantee retrieving the most helpful knowledge. A larger knowledge base includes more diverse scenarios but we might also retrieve less relevant knowledge. However, we observed that these cases are rare and do not significantly impact overall performance.
>
> > *What is the result when the user provides an ambiguous answer when the system seeks clarification?*
>
> We do consider scenarios where the user could have multiple intents. For example, if the user asks for "bring me a soda" when there are both Coke and Sprite, selecting either option is considered a success. This differs from "bring me that soda," where the user has a specific intent regarding which soda.

---

> > ### Comment · Reviewer_Gr1N · 2024-08-14
> > **Response**
> >
> > Thank you providing additional clarity to my questions and comments. Also, thank you for providing the additional plot on the size of the knowledge bag. This clarifies my understanding of the sytem and confidence in my rating.

---

> > > ### Author Response · Authors · 2024-08-14
> > >
> > > Thanks for your response! We appreciate your review and feedback!

---

### Official Review · Reviewer_QU2A · 2024-07-25

**Soundness:** 3
**Presentation:** 4
**Contribution:** 3
**Rating:** 7
**Confidence:** 3

**Summary:**

This paper proposes a new method of using LLM to do task planning. The key innovation is a retrieval-augmented generation (RAG) where the LLM retrieves few-shot introspective reasoning examples from a knowledge base that contains examples with supervised labels. The authors integrate the RAG into LLM to either (1) make it directly predict the best plan, and (2) make it use conformal prediction to predict the best plan. Results have shown that the proposed method, LLM + introspective reasoning + direct prediction outperformed all baselines. And LLM + introspective reasoning + conformal prediction outperformed all baselines with conformal predictions. However, LLM + introspective reasoning + direct prediction outperformed LLM + introspective reasoning + conformal prediction, which is an open question for future work.

**Strengths:**

- The method is sound where the main message is very simple - introspective reasoning based on a dataset with ground truth labels is helpful for LLM planning.
- The writing is very clear.
- This paper also proposes new evaluation metrics, which more comprehensively measures planners' performance.
- The experiment is rich with 3 problem domains with various baseline methods, including the non-conformal-prediction-based and the conformal-prediction-based.
- The result has shown the benefit of incorporating introspective reasoning.

**Weaknesses:**

Besides the limitation discussed in the end of the paper, there are two other potential limitations:
- The proposed method relies on a knowledge base with human supervised labels, which may require significant human efforts to construct. It might be useful to discuss the cost of such knowledge base.
- The usage of both introspective planning and conformal prediction might increase the computation load for inference. It might be useful to discuss the increased computation.

**Questions:**

- In Eq.2, why choosing (N+1)(1-ε)/N?
- Line 135-141 highlights the difference of the proposed work vs [31]. Just to make sure I understand it correctly, the key difference is incorporating `k` in Eq.3?

**Limitations:**

Limitations are discussed in the end of the paper.

---

> ### Author Rebuttal · Authors · 2024-08-07
>
> We thank the reviewer for the positive feedback, acknowledging that our method is sound, the writing is clear, and the proposed metrics comprehensively measure the planner's performance. We also thank you for recognizing the valuable experimental results we have presented.
>
> > *The proposed method relies on a knowledge base with human supervised labels, which may require significant human efforts to construct. It might be useful to discuss the cost of such knowledge base.*
>
> In our approach, humans only need to provide labels (instead of explanation or reasoning), and we leverage the LLM to generate the reasoning based on these labels, which are then stored as knowledge. This significantly reduces the human effort required. Labeling the knowledge base data with $400$ examples takes approximately $3$ hours.
>
> > *The usage of both introspective planning and conformal prediction might increase the computation load for inference. It might be useful to discuss the increased computation.*
>
> Using introspective planning does indeed increase the computation load for inference. We discussed the cost and efficiency in Appendix D. For our experiments on Mobile Manipulation with a calibration set size of 400, a test set size of 400, and a knowledge base size of 400, KnowNo requires 35k completion tokens and 376k prompt tokens in total. Our approach requires three times the completion tokens and twice the prompt tokens. Consequently, the total API cost for KnowNo is $4.8, while ours is approximately $9.8. Despite the increased cost, our method significantly improves the Exact Set Rate, with KnowNo achieving only 37% compared to our method's 94.5%.
>
> > *In Eq.2, why choosing (N+1)(1-ε)/N?*
>
> To ensure the prediction set meets the desired coverage level with the specified confidence $1 - \epsilon$, we use the quantile threshold defined as follows:
>
>
> $\hat{q} = \text{Quantile}(s_1, ..., s_N ; \frac{\lceil (N+1)(1-\epsilon) \rceil}{N})$
>
> This means $\hat{q}$ is the $\lceil (N+1)(1-\epsilon) \rceil$-th smallest value among the nonconformity scores $s_1, \ldots, s_N$. When we compute $\hat{q}$, we are effectively selecting this specific quantile to ensure the desired coverage. For a new task $x_{\text{test}}$, we include all labels $y$ in the prediction set $\hat{\mathcal{G}}_{\text{test}}$ such that:
>
> $\hat{f}(y \mid x_{\text{test}}, \mathcal{C}_{\text{test}}, k) \geq 1 - \hat{q}$
>
> This criterion ensures that the prediction set $\hat{\mathcal{G}}_{\text{test}}$ includes all labels $y$ where the confidence level $\hat{f}$ meets or exceeds $1 - \hat{q}$. With this construction, we achieve the following coverage guarantee:
>
> $\mathbb{P}(z_{\text{test}} \in \hat{\mathcal{G}}_{\text{test}}) \geq 1 - \epsilon$
>
> This guarantees that the true intent $z_{\text{test}}$ is included in the prediction set $\hat{\mathcal{G}}_{\text{test}}$ with a probability of at least $1 - \epsilon$.
>
> > *Line 135-141 highlights the difference of the proposed work vs [31]. Just to make sure I understand it correctly, the key difference is incorporating k in Eq.3?*
>
> Yes, the key difference compared to [31] is the incorporation of the introspective planning rationale $k_i$ in our framework. This rationale is generated through the process described in lines 74-85, and it significantly improves planner performance in both compliance and safety. It also refines LLM uncertainty and provides a tighter bound of guarantee.

---

> > ### Comment · Reviewer_QU2A · 2024-08-12
> > **keeping my score**
> >
> > Thank you for the detailed explanation. After reading other reviews, it seems that one main weakness is whether integrating introspective reasoning is sufficiently novel. As I am not familiar with the literature in LLM for planning, I will defer to other reviewers about evaluating the contribution. For now, I am inclined to keep the score.

---

> > > ### Author Response · Authors · 2024-08-12
> > >
> > > We really appreciate your response and efforts in the review process! We have included additional notes regarding the novelty of introspective planning in the replies of other reviewers, and I hope they can be helpful as well.

---

### Author Rebuttal · Authors · 2024-08-07

We sincerely appreciate the time and effort that all reviewers and area chairs have dedicated to providing valuable feedback and constructive advice on our manuscript. We are encouraged by the consensus among reviewers on the importance of guiding language agents to reason about their own uncertainty and ensure safety during decision-making. Detailed responses to each review are provided in the sections below. The attached PDF contains a figure that evaluates the influence of the knowledge base size on a few additional metrics (OAR, OSR, and NCR), addressing the comment from Reviewer Gr1N.

Here we listed all the required references that we used in the rebuttal where [1, 19, 31] are in the paper with the same index.

[1] Ahn, Michael, et al. "Do as I can, not as I say: Grounding language in robotic affordances." arXiv preprint arXiv:2204.01691 (2022).

[19] David B. Leake. Introspective Learning and Reasoning, pages 1638–1640. Springer US, Boston, 390 MA, 2012.

[31] Ren, Allen Z., et al. "Robots that ask for help: Uncertainty alignment for large language model planners." arXiv preprint arXiv:2307.01928 (2023).

[2] Lyu, Chenyang, Minghao Wu, and Alham Fikri Aji. "Beyond probabilities: Unveiling the misalignment in evaluating large language models." arXiv preprint arXiv:2402.13887 (2024).

[3] Wang, Xinpeng, et al. "" My Answer is C": First-Token Probabilities Do Not Match Text Answers in Instruction-Tuned Language Models." arXiv preprint arXiv:2402.14499 (2024).

---

> ### Author Response · Authors · 2024-08-12
>
> Dear Reviewers,
>
> We sincerely appreciate the time and effort you have dedicated to reviewing our paper. As the discussion period approaches its conclusion, we kindly remind you of the upcoming deadline. Please feel free to raise any points or questions that may require further clarification.
>
> Your insights and feedback are invaluable to us. Thank you once again for your thoughtful review.

---

### Decision · Program_Chairs · 2024-09-25

**Decision:**

Accept (poster)

**Comment:**

This paper introduces a LLM-based task planning method that aims to better understand uncertainties in plans and user intents. Initially, some reviewers raised concerns about its technical novelty, particularly compared to prior work using Chain of Thought and conformal prediction. However, these concerns were largely resolved through the author-reviewer discussion. This led to a general consensus among reviewers to accept this paper. One reviewer expressed very negative views, but the authors' response appears to have adequately addressed these concerns (although the reviewer did not provide a follow-up reply).

While this paper has been recommended for acceptance, the authors are advised to revisit the reviewers' comments to improve the clarity of their writing. Particularly, the authors should provide clearer explanations of the term "introspective planning" and refining their claims (e.g. tighter confidence bound).